# Extracting the Ultimate New Soliton Solutions of Some Nonlinear Time Fractional PDEs via the Conformable Fractional Derivative

Md Ashik Iqbal [1], Abdul Hamid Ganie [2], Md Mamun Miah [3,4,*] and Mohamed S. Osman [5,*]

[1] Department of Mathematics & Physics, Khulna Agricultural University, Khulna 9100, Bangladesh; ashikiqbalmath@gmail.com

[2] Department of Basic Science, College of Science and Theoretical Studies, Saudi Electronic University, Riyadh 11673, Saudi Arabia; a.ganie@seu.edu.sa

[3] Division of Mathematical and Physical Sciences, Kanazawa University, Kanazawa 9201192, Japan

[4] Department of Mathematics, Khulna University of Engineering & Technology, Khulna 9203, Bangladesh

[5] Department of Mathematics, Faculty of Science, Cairo University, Giza 12613, Egypt

* Correspondence: mamun0954@gmail.com (M.M.M.); mofatzi@cu.edu.eg or mofatzi@sci.cu.edu.eg (M.S.O.)

**Abstract:** Nonlinear fractional-order differential equations have an important role in various branches of applied science and fractional engineering. This research paper shows the practical application of three such fractional mathematical models, which are the time-fractional Klein–Gordon equation (KGE), the time-fractional Sharma–Tasso–Olever equation (STOE), and the time-fractional Clannish Random Walker's Parabolic equation (CRWPE). These models were investigated by using an expansion method for extracting new soliton solutions. Two types of results were found: one was trigonometric and the other one was an exponential form. For a profound explanation of the physical phenomena of the studied fractional models, some results were graphed in 2D, 3D, and contour plots by imposing the distinctive results for some parameters under the oblige conditions. From the numerical investigation, it was noticed that the obtained results referred smooth kink-shaped soliton, ant-kink-shaped soliton, bright kink-shaped soliton, singular periodic solution, and multiple singular periodic solutions. The results also showed that the amplitude of the wave augmented with the pulsation in time, which derived the order of time fractional coefficient, remarkably enhanced the wave propagation, and influenced the nonlinearity impacts.

**Keywords:** the $\left(\frac{G'}{G'+G+A}\right)$-expansion method; the conformable fractional derivative; the time-fractional Klein–Gordon equation; the time-fractional Sharma–Tasso–Olever equation; the time-fractional Clannish Random Walker's Parabolic equation

## 1. Introduction

The mysteries of various complex phenomena occurring in nature can be explained by providing fractional mathematical models with the help of fractional-order derivatives and by providing exact soliton solutions of these models. Studying such mathematical models has a firm place in various fields of engineering, modern medical science, and physical science. As an evolution of integer-order perception, fractional calculus is valuable for modeling real-world occurrences with complex dynamics [1,2]. Many studies have shown that fractional-order differential equations can successfully express complex forcible characteristics [3,4]. Moreover, by applying fractional derivatives, we can explain the reliability of the memory effects, called a fundamental approach, to various real-world occurrences [5]. Over the past few years, numerous interpretations of fractional derivatives have been proposed due to the rewards provided by giving models of real-world phenomena. It has been shown that fractional differential equations, especially fractional-order partial differential equations (PDEs), have made an indispensable contribution in many scientific and engineering sectors. The use of such fractional PDEs in fields like theoretical physics,

plasma physics, computational biology, applied physics, biochemistry, signal processing, systems identification, electronic communication, including blogs and Facebook, electromagnetism, electrochemistry, nanotechnology, nonlinear optics, fluid mechanics, control theory, finance, fractional dynamics, etc., is continuously increasing [6,7]. Since fractional operators have a single kernel in the classical frame, many nonlinear complex phenomena cannot be exactly characterized by traditional illustrations. Therefore, new fractional derivatives with nonsingular kernels were developed to investigate nonlocal dynamics, such as Atangana–Baleanu–Caputo [8,9] and fractal-fractional operator kernels [10].

Moreover, the applications of the spectral approach in the procedure for obtaining solutions to the space–time non-integer order reaction–diffusion situation can be noticed in [11,12]. We note that fractional-order derivatives [13,14] are difficult to deal with analytically, primarily those that describe real-world operations, and investigators sometimes must depend on the numerical approach to solve these equations. Many investigators have used various operators of fractional derivatives like Caputo–Fabrizio [15,16], Riemann–Liouville [17,18], conformable fractional derivatives [19,20], etc., in fractional-order PDEs of any system. One of the most familiar fractional-order PDE is the nonlinear KGE [21], which has widespread implementations in condensed-type matter physics, nonlinear optics, quantum mechanics, etc., and is also suitable for modeling real-world occurrences. The second one is the nonlinear STOE [22], which is essential in applied science and engineering. The nonlinear STOE explains the factual naturalistic model of fusion and fission, which Kupershmidt introduces as the idea of dark equations in model equations [23]. The third one is the nonlinear CRWPE [24], which is vital in understanding the quantitative and qualitative features of various natural phenomena. For example, the nonlinear oscillation of an earthquake can be modeled by the nonlinear CRWPE. Our cited model equations can gain complete shock-wave or topological-wave solutions. The fast progress of systematic methods for getting the solution to fractional PDEs has been of considerable help in addressing physical-world problems of a complicated nature. Consequently, researchers have focused on investigating fractional-order calculus and detecting exact and methodical techniques for finding the perfect solutions for fractional PDEs. Recently, many researchers have investigated these types of equations by applying different methods, such as the Generalized Kudryashov method [25,26], the residual-power-series method [27,28], the exp-function method [29,30], the long-wave method [31], the variational iteration method [32,33], the extended direct algebraic method [34,35], the sine-Gordon expansion approach [36], the Jacobi elliptic function method [37], the Sarder sub-equation method [38], the $\left( \frac{G'}{G}, \frac{1}{G} \right)$-expansion method [39–41], and many other techniques. Now, there is a more well-organized method called the $\left( \frac{G'}{G'+G+A} \right)$-expansion method [42,43] to solve nonlinear fractional-order PDEs.

This research article's prime goal is to find the exact solitary-wave solutions of our proposed three model equations through this simple mathematical expansion method.

The three proposed nonlinear time-fractional PDEs are given below:

- The nonlinear KGE is [44,45]

$$\frac{\partial^{2\beta} v}{\partial t^{2\beta}} - \frac{\partial^2 v}{\partial x^2} - k_1 v - k_2 v^3 = 0; \ t > 0, \ 0 < \beta \leq 1, \tag{1}$$

where $k_1$ and $k_2$ are any constants and $\beta$ represents the time derivative of fractional order.

- The nonlinear STOE is [46,47]

$$\frac{\partial^{\beta} v}{\partial t^{\beta}} + 3d \left( \frac{\partial v}{\partial x} \right)^2 + 3dv^2 \frac{\partial v}{\partial x} + 3dv \frac{\partial^2 v}{\partial x^2} + d \frac{\partial^3 v}{\partial x^3} = 0; \ t > 0, \ 0 < \beta \leq 1. \tag{2}$$

Here, $d$ is an arbitrary constant.

- The nonlinear CRWPE is [48,49]

$$\frac{\partial^\beta v}{\partial t^\beta} - \frac{\partial v}{\partial x} + 2v\frac{\partial v}{\partial x} + \frac{\partial^2 v}{\partial x^2} = 0, \ t > 0, \ 0 < \beta \le 1 \ \text{and} \ x \in \mathbb{R}. \tag{3}$$

The configuration of this article is split into six parts as follows: In Section 2, we give a short explanation of the conformable fractional derivative with some characteristics. The general study of our proposed expansion method is described in Section 3. The analytical implementation of our proposed technique to extract the exact solitary wave solutions of the KGE, the STOE, and the CRWPE is in Section 4. In Section 5, the outcomes and graphical explanations are given. Ultimately, the conclusion is conferred in Section 6.

## 2. Definition of Conformable Fractional Derivative and Its Characteristics

The generally accepted notion of the fractional-order integral differential operator is defined as follows [50–54]:

$$_aD_x^\beta = \begin{cases} \frac{d^\beta}{dx^\beta}, & f(\beta) > 0 \\ 1, & f(\beta) = 0 \\ \int_t^x d\varphi, & f(\beta) < 0 \end{cases}, \tag{4}$$

where $\beta$ is the fractional order and $f(\beta)$ implies the real part of $\beta$, and $a$ is the lower limit of the operation, which is constant. On the contrary, the upper limit $x$ varies with $x > a$.

Recently, some researchers have developed definitions of some essential fractional derivatives, such as the conformable fractional derivative, the Riemann–Liouville derivative, the modified Riemann–Liouville derivative, the Caputo derivative, the generalized Riemann–Liouville–Caputo derivative, the Caputo–Fabrizio derivative, Atangana–Baleanu derivative, etc. Here, the characteristics and definitions of a simple fractional-order derivative described by Khalil et al. [55] are given, called the conformable fractional derivative. In our article, we used the $\left(\frac{G'}{G'+G+A}\right)$-expansion method for solving our proposed three nonlinear model equations in the sense of the conformable fractional derivative [56,57].

If $F: R^+ \to R$ is a continuous function, then the definition of the conformable fractional derivative of order $\beta$ is written as

$$\frac{\partial^\beta F}{\partial t^\beta} = \lim_{\delta \to 0^+} \frac{F(\delta t^{1-\beta} + t) - F(t)}{\delta}, \tag{5}$$

where $t$ is positive and $\delta \in (0, 1)$.

**Characteristics:** Consider the two $\beta$-differentible functions $F$ and $G$; then,

$$\text{(i)} \ \frac{\partial^\beta t^\alpha}{\partial t^\beta} = qt^{\alpha-\beta}, \ \alpha \in \mathbb{R}. \tag{6}$$

$$\text{(ii)} \ \frac{\partial^\beta}{\partial t^\beta}(\text{constant}) = 0 \tag{7}$$

$$\text{(iii)} \ \frac{\partial^\beta}{\partial t^\beta}(r_1 F + r_2 G) = r_1\frac{\partial^\beta F}{\partial t^\beta} + r_2\frac{\partial^\beta G}{\partial t^\beta}, \tag{8}$$

where $r_1, r_2$ are real constants.

$$\text{(iv)} \ \frac{\partial^\beta}{\partial t^\beta}(FG) = F\frac{\partial^\beta G}{\partial t^\beta} + G\frac{\partial^\beta F}{\partial t^\beta}. \tag{9}$$

$$\text{(v)} \ \frac{\partial^\beta}{\partial t^\beta}\left(\frac{F}{G}\right) = \frac{G\frac{\partial^\beta F}{\partial t^\beta} - F\frac{\partial^\beta G}{\partial t^\beta}}{G^2}. \tag{10}$$

$$\text{(vi)} \ \frac{\partial^\beta F}{\partial t^\beta} = t^{(1-\beta)}\frac{dF}{dt}, \ \text{when } F \text{ is differentiable.} \tag{11}$$

(vii) If $F$ and $G$ are $\beta$-differentiable function of $t$ in the domain $(0, \infty)$ and $G(t) \neq 0$ then,

$$\frac{\partial^\beta}{\partial t^\beta}(FoG)(t) = (T_\beta F)(G(t))(T_\beta G)(t)G(t)^{\beta-1}. \tag{12}$$

## 3. Discussion of the Expansion Method

The main steps of the $\left(\frac{G'}{G'+G+A}\right)$-expansion method are given in detail in this portion. Consider the following nonlinear PDEs as

$$\psi\left(v, \ \partial_x^\beta v, \ \partial_t^\beta v, \ \partial_x^\beta \partial_x^\beta v, \ \partial_x^\beta \partial_t^\beta v, \ \partial_t^\beta \partial_t^\beta v \ldots\right) = 0. \tag{13}$$

where polynomial $\psi$ is a function of $v = v(x, t)$ and its derivatives. Now, the mechanisms of this method are given periodically.

**Mechanism I:** Using the wave conversion technique, we first wrote all the variables in the proposed PDEs as a linear combination of a unique variable as follows:

$$v(x, \ t) = v(\rho); \ \rho = mx - \frac{\mu}{\beta}t^\beta. \tag{14}$$

Here, the coefficient of space derivative $m$ is called the wave number, and the coefficient of time derivative $\mu$ is called the wave velocity and has constant values. Applying the wave conversion technique from Equation (14) to (13), a simple ordinary differential equation (ODE) can be written as

$$\chi\left(v, \ v', \ v'', \ v''', \ \ldots\right) = 0. \tag{15}$$

Here, $v' = \frac{dv}{d\rho}$, $v'' = \frac{d^2v}{d\rho^2}$, $v''' = \frac{d^3v}{d\rho^3}$ etc.

**Mechanism II:** Now, we must consider an equation like the following form, which will be the solution of Equation (15)

$$v(\rho) = \sum_{r=0}^M a_r \left(\frac{G'}{G' + G + A}\right)^r. \tag{16}$$

where $a_r$ are the coefficients of the polynomial $\left(\frac{G'}{G'+G+A}\right)^r, r = 0, 1, 2, \ldots M$ and consider the function $G(\rho)$ that satisfies the following ODE:

$$G'' + HG' + RG + RA = 0. \tag{17}$$

Now, the value of balance number $M$ will be determined by applying the homogeneous balance technique between the supreme-order derivative and the supreme-degree convective terms in Equation (15). By assigning the value of the balance number $M$ in Equation (16) and simplifying the resultant, we performed the derivative with respect to $\rho$ as often as was needed, then set these derivatives in Equation (15).

**Mechanism III:** From Equation (15), the coefficients of the successive power of $\left(\frac{G'}{G'+G+A}\right)$ are taken to be zero, which provides some equations in terms of $H$, $R$, $A$, $\mu$, $m$, and $a_r(r = 0, 1, 2, \ldots M)$. After solving these equations via the program Mathematica, the values of $a_r(r = 0, 1, 2, \ldots M)$ will be found. Now, we find the exponential and trigonometric function solutions for the positive and negative values of the discriminant of the auxiliary equation in Equation (17). Then, we put the values of the solution $G$ in the term $\left(\frac{G'}{G'+G+A}\right)$ and finally, for the values of $\left(\frac{G'}{G'+G+A}\right)$, $a_r$, and $\rho$, we have the ultimate solution of Equation (13).

## 4. Applications and Discussions

*4.1. Investigation of the KGE*

By setting the following transformation in Equation (1), we obtain the following second equation:

$$v(x,\ t) = v(\rho),\ \rho = mx - \frac{\mu}{\beta}t^{\beta}. \tag{18}$$

$$\left(\mu^2 - m^2\right)\frac{d^2v}{d\rho^2} - k_1v - k_2v^3 = 0. \tag{19}$$

By applying the homogenous balance rule between the terms $v^3$ and $\frac{d^2v}{d\rho^2}$ in Equation (19), the balance number $M = 1$. Equation (16) moves as

$$v(\rho) = a_0 + a_1\left(\frac{G'}{G' + G + A}\right). \tag{20}$$

Inserting Equation (20) along with Equation (17) into Equation (19) and then placing the coefficients of $\left(\frac{G'}{G'+G+A}\right)^r$; $(r = 0,\ 1,\ 2,\ldots M)$ to zero into the resultant, a set of equations in terms of $a_0$, $a_1$, $m$, $\mu$, $H$, $R$, $k_1$ and $k_2$ will be obtained as follows:

$$2\mu^2a_1R^2 - 2m^2a_1R^2 - \mu^2a_1HR + m^2a_1HR + k_2a_0^3 + k_1a_0 = 0,$$

$$6m^2a_1HR - 6\mu^2a_1HR - 2m^2a_1R + \mu^2a_1H^2 + 6\mu^2a_1R^2 + 2\mu^2a_1R -$$
$$m^2a_1H^2 - 6m^2a_1R^2 - k_1a_1 - 3k_2a_0^2a_1 = 0,$$

$$9\mu^2a_1HR - 3k_2a_0a_1^2 - 6\mu^2a_1R^2 - 6\mu^2a_1R - 9m^2a_1HR - 3m^2a_1H -$$
$$3\mu^2a_1H^2 + 3\mu^2a_1H + 3m^2a_1H^2 + 6m^2a_1R + 6m^2a_1R^2 = 0,$$

$$4\mu^2a_1HR - 4m^2a_1HR - 4\mu^2a_1R + 2m^2a_1H^2 + k_2a_1^3 - 2\mu^2a_1R^2 +$$
$$2m^2a_1R^2 - 4m^2a_1H + 4m^2a_1R + 4\mu^2a_1H - 2\mu^2a_1H^2 + 2m^2a_1 -$$
$$2\mu^2a_1 = 0.$$

By solving the above system via the software Mathematica 11, we obtain the following set of values:

$$a_0 = a_0,\ a_1 = \frac{2a_0(H-R-1)}{2R-H},\ m = m,\ \mu = \mu,\ k_1 = \frac{1}{2}m^2H^2 -$$
$$\frac{1}{2}\mu^2H^2 + 2\mu^2R - 2m^2R,\ k_2 = \frac{(H-2R)^2\left(\mu^2-m^2\right)}{2a_0^2}.$$

Now, the soliton solutions to the KGE are given for two occurrences as follows:
**Occurrence 1.** For $D = H^2 - 4R > 0$,

$$v(\rho) = a_0 + \frac{2a_0(H-R-1)}{2R-H} \times \left[\frac{c_1\left(H+\sqrt{D}\right) + c_2\left(H-\sqrt{D}\right)e^{\sqrt{D}\rho}}{c_1\left(H+\sqrt{D}-2\right) + c_2\left(H-\sqrt{D}-2\right)e^{\sqrt{D}\rho}}\right]. \tag{21}$$

Using the wave transformation from Equation (18), we can write the soliton solution of Equation (1) as

$$v(x,\ t) = a_0 + \frac{2a_0(H-R-1)}{2R-H} \times \left[\frac{c_1\left(H+\sqrt{D}\right) + c_2\left(H-\sqrt{D}\right)e^{\sqrt{D}[mx-\frac{\mu}{\beta}t^{\beta}]}}{c_1\left(H+\sqrt{D}-2\right) + c_2\left(H-\sqrt{D}-2\right)e^{\sqrt{D}[mx-\frac{\mu}{\beta}t^{\beta}]}}\right]. \tag{22}$$

**Occurrence 2.** For $D = H^2 - 4R < 0$,

$$v(\rho) = a_0 + \frac{2a_0(H-R-1)}{2R-H} \times$$
$$\left[\frac{(Hc_2+c_1\sqrt{-D})\sin\left(\frac{\sqrt{-D}}{2}\rho\right) + (Hc_1-c_2\sqrt{-D})\cos\left(\frac{\sqrt{-D}}{2}\rho\right)}{((H-2)c_2+c_1\sqrt{-D})\sin\left(\frac{\sqrt{-D}}{2}\rho\right) + ((H-2)c_1-c_2\sqrt{-D})\cos\left(\frac{\sqrt{-D}}{2}\rho\right)}\right]. \tag{23}$$

Using the wave transformation from Equation (18), the other soliton solution of Equation (1) is written as

$$v(x,\ t) = a_0 + \frac{2a_0(H-R-1)}{2R-H} \times$$

$$\left[ \frac{(Hc_2+c_1\sqrt{-D})sin\left(\frac{\sqrt{-D}}{2}[mx-\frac{\mu}{\beta}t^\beta]\right)+(Hc_1-c_2\sqrt{-D})cos\left(\frac{\sqrt{-D}}{2}[mx-\frac{\mu}{\beta}t^\beta]\right)}{((H-2)c_2+c_1\sqrt{-D})sin\left(\frac{\sqrt{-D}}{2}[mx-\frac{\mu}{\beta}t^\beta]\right)+((H-2)c_1-c_2\sqrt{-D})cos\left(\frac{\sqrt{-D}}{2}[mx-\frac{\mu}{\beta}t^\beta]\right)} \right]. \tag{24}$$

### 4.2. Investigation of the STOE

By setting the following transformation in Equation (2), we obtain the following second equation:

$$v(x,\ t) = v(\rho),\ \rho = x - \frac{\mu}{\beta}t^\beta. \tag{25}$$

$$d\frac{d^3v}{d\rho^3} + 3d\ v\frac{d^2v}{d\rho^2} + 3d\left(\frac{dv}{d\rho}\right)^2 + 3d\ v^2\frac{dv}{d\rho} - \mu\frac{dv}{d\rho} = 0. \tag{26}$$

Integrating the above equation, we obtain

$$N - \mu v + 3d\ v\frac{dv}{d\rho} + d\ v^3 + d\ \frac{d^2v}{d\rho^2} = 0, \tag{27}$$

where $N$ is the integration constant. By applying the balance rule between the terms $v^3$ and $\frac{d^2v}{d\rho^2}$ in Equation (27), the balance number, $M = 1$. Equation (16) moves as

$$v(\rho) = a_0 + a_1\left(\frac{G'}{G'+G+A}\right). \tag{28}$$

Inserting Equation (28) along with Equation (17) into (27) and then placing the coefficients of $\left(\frac{G'}{G'+G+A}\right)^r$; $(r = 0,\ 1,\ 2, \ldots M)$ to zero into the resultant, a set of equations in terms of $a_0$, $a_1$, $\mu$, $H$, $R$, $N$, and $d$ will be obtained, and after solving the obtaining set of equations via the software Mathematica, two sets of results are attained as follows.

Set 1:

$$a_0 = a_0,\ a_1 = R - H + 1,\ d = d,\ N = -6da_0HR + 2da_0R + 6da_0R^2 + dH^2R - dHR -$$
$$3dHR^2 + 2dR^2 + 2dR^3 + 6da_0^2R + 2da_0^3 - 3da_0^2H + a_0dH^2,\ \mu = -3dHR - dR +$$
$$6da_0R + 3dR^2 + 3da_0^2 - 3da_0H + dH^2.$$

Set 2:

$$a_0 = H - 2R,\ a_1 = 2R - 2H + 2,\ N = 0,\ d = d,\ \mu = -4dR + dH^2.$$

For set 1, the soliton solutions to the STOE are given for two occurrences as follows:
**Occurrence 1.** For $D = H^2 - 4R > 0$,

$$v(\rho) = a_0 + (R - H + 1) \times \left[ \frac{c_1\left(H + \sqrt{D}\right) + c_2\left(H - \sqrt{D}\right)e^{\sqrt{D}\rho}}{c_1\left(H + \sqrt{D} - 2\right) + c_2\left(H - \sqrt{D} - 2\right)e^{\sqrt{D}\rho}} \right]. \tag{29}$$

Using the wave transformation from Equation (25), we can write the soliton solution of Equation (2) as

$$v(x,\ t) = a_0 + (R - H + 1) \times \left[ \frac{c_1\left(H + \sqrt{D}\right) + c_2\left(H - \sqrt{D}\right)e^{\sqrt{D}[x-\frac{\mu}{\beta}t^\beta]}}{c_1\left(H + \sqrt{D} - 2\right) + c_2\left(H - \sqrt{D} - 2\right)e^{\sqrt{D}[x-\frac{\mu}{\beta}t^\beta]}} \right], \tag{30}$$

where $\mu = -3dHR - dR + 6da_0R + 3dR^2 + 3da_0^2 - 3da_0H + dH^2.$

**Occurrence 2.** For $D = H^2 - 4R < 0$,

$$v(\rho) = a_0 + (R - H + 1) \times \left[ \frac{(Hc_2 + c_1\sqrt{-D})\sin\left(\frac{\sqrt{-D}}{2}\rho\right) + (Hc_1 - c_2\sqrt{-D})\cos\left(\frac{\sqrt{-D}}{2}\rho\right)}{((H-2)c_2 + c_1\sqrt{-D})\sin\left(\frac{\sqrt{-D}}{2}\rho\right) + ((H-2)c_1 - c_2\sqrt{-D})\cos\left(\frac{\sqrt{-D}}{2}\rho\right)} \right]. \quad (31)$$

Again, using the wave transformation from Equation (25), the soliton solution of Equation (2) is written as

$$v(x, t) = a_0 + (R - H + 1) \times \left[ \frac{(Hc_2 + c_1\sqrt{-D})\sin\left(\frac{\sqrt{-D}}{2}[x - \frac{\mu}{\beta}t^\beta]\right) + (Hc_1 - c_2\sqrt{-D})\cos\left(\frac{\sqrt{-D}}{2}[x - \frac{\mu}{\beta}t^\beta]\right)}{((H-2)c_2 + c_1\sqrt{-D})\sin\left(\frac{\sqrt{-D}}{2}[x - \frac{\mu}{\beta}t^\beta]\right) + ((H-2)c_1 - c_2\sqrt{-D})\cos\left(\frac{\sqrt{-D}}{2}[x - \frac{\mu}{\beta}t^\beta]\right)} \right], \quad (32)$$

where $\mu = -3dHR - dR + 6da_0R + 3dR^2 + 3da_0^2 - 3da_0H + dH^2$.

For set 2, the soliton solutions to the STOE are given for two occurrences as follows,

**Occurrence 1.** For $D = H^2 - 4R > 0$,

$$v(\rho) = (H - 2R) + (2R - 2H + 2) \times \left[ \frac{c_1\left(H + \sqrt{D}\right) + c_2\left(H - \sqrt{D}\right)e^{\sqrt{D}\rho}}{c_1\left(H + \sqrt{D} - 2\right) + c_2\left(H - \sqrt{D} - 2\right)e^{\sqrt{D}\rho}} \right]. \quad (33)$$

Using the wave transformation from Equation (25), the other soliton solution of Equation (2) is written as

$$v(x, t) = (H - 2R) + (2R - 2H$$
$$+2) \times \left[ \frac{c_1\left(H+\sqrt{D}\right)+c_2\left(H-\sqrt{D}\right)e^{\sqrt{D}[x - \frac{d(H^2-4R)}{\beta}t^\beta]}}{c_1\left(H+\sqrt{D}-2\right)+c_2\left(H-\sqrt{D}-2\right)e^{\sqrt{D}[x - \frac{d(H^2-4R)}{\beta}t^\beta]}} \right]. \quad (34)$$

**Occurrence 2.** For $D = H^2 - 4R < 0$,

$$v(\rho) = (H - 2R) + (2R - 2H + 2) \times \left[ \frac{(Hc_2 + c_1\sqrt{-D})\sin\left(\frac{\sqrt{-D}}{2}\rho\right) + (Hc_1 - c_2\sqrt{-D})\cos\left(\frac{\sqrt{-D}}{2}\rho\right)}{((H-2)c_2 + c_1\sqrt{-D})\sin\left(\frac{\sqrt{-D}}{2}\rho\right) + ((H-2)c_1 - c_2\sqrt{-D})\cos\left(\frac{\sqrt{-D}}{2}\rho\right)} \right]. \quad (35)$$

Again, using the wave transformation from Equation (25), the other soliton solution of Equation (2) is written as

$$v(x, t) = (H - 2R) + (2R - 2H + 2) \times$$
$$\left[ \frac{(Hc_2+c_1\sqrt{-D})\sin\left(\frac{\sqrt{-D}}{2}\left[x - \frac{d(H^2-4R)}{\beta}t^\beta\right]\right) + (Hc_1-c_2\sqrt{-D})\cos\left(\frac{\sqrt{-D}}{2}\left[x - \frac{d(H^2-4R)}{\beta}t^\beta\right]\right)}{((H-2)c_2+c_1\sqrt{-D})\sin\left(\frac{\sqrt{-D}}{2}\left[x - \frac{d(H^2-4R)}{\beta}t^\beta\right]\right) + ((H-2)c_1-c_2\sqrt{-D})\cos\left(\frac{\sqrt{-D}}{2}\left[x - \frac{d(H^2-4R)}{\beta}t^\beta\right]\right)} \right]. \quad (36)$$

### 4.3. Investigation of the CRWPE

By setting the following transformation in Equation (3), we obtain the following second equation:

$$v(x, t) = v(\rho), \quad \rho = px - \frac{\eta}{\beta}t^\beta. \quad (37)$$

$$-(p + \eta)\frac{dv}{d\rho} + 2pv\frac{dv}{d\rho} + p^2\frac{d^2v}{d\rho^2} = 0. \quad (38)$$

Integrating the above equation, we obtain

$$-(p + \eta)v + pv^2 + p^2\frac{dv}{d\rho} + \rho_0 = 0, \quad (39)$$

where $\rho_0$ is the integration constant. By applying the balance rule between terms $v^2$ and $\frac{dv}{d\rho}$ in Equation (39), the balance number, $M = 1$. Equation (16) moves as

$$v(\rho) = a_0 + a_1 \left( \frac{G'}{G' + G + A} \right). \tag{40}$$

Inserting Equation (40) along with Equation (17) into Equation (39) and then placing the coefficients of $\left( \frac{G'}{G'+G+A} \right)^r$, $(r = 0, 1, 2, \ldots M)$ to zero into the resultant, a set of equations in terms of $a_0, a_1, \eta, H, R, p$, and $\rho_0$ will be obtained, and after solving the obtaining set of equations via the software Mathematica, the following set of particular values are obtained as follows:

$$a_0 = a_0, \quad a_1 = p(R - H + 1), \quad p = p, \quad \eta = -p + 2p^2 R + 2pa_0 - p^2 H,$$

$$\rho_0 = 2a_0 p^2 R + pa_0^2 - a_0 p^2 H + p^3 R^2 - p^3 HR + p^3 R.$$

For the above set, the soliton solutions to the nonlinear CRWP equation are given for two occurrences as follows:

**Occurrence 1.** For $D = H^2 - 4R > 0$,

$$v(\rho) = a_0 + p(R - H + 1) \times \left[ \frac{c_1 \left( H + \sqrt{D} \right) + c_2 \left( H - \sqrt{D} \right) e^{\sqrt{D}\rho}}{c_1 \left( H + \sqrt{D} - 2 \right) + c_2 \left( H - \sqrt{D} - 2 \right) e^{\sqrt{D}\rho}} \right]. \tag{41}$$

Using the wave transformation from Equation (37), the soliton solution of Equation (3) is written as

$$v(x, t) = a_0 + p(R - H + 1) \times \left[ \frac{c_1 \left( H + \sqrt{D} \right) + c_2 \left( H - \sqrt{D} \right) e^{\sqrt{D}[px - \frac{\eta}{\beta} t^\beta]}}{c_1 \left( H + \sqrt{D} - 2 \right) + c_2 \left( H - \sqrt{D} - 2 \right) e^{\sqrt{D}[px - \frac{\eta}{\beta} t^\beta]}} \right], \tag{42}$$

where $\eta = -p + 2p^2 R + 2pa_0 - p^2 H$.

**Occurrence 2.** For $D = H^2 - 4R < 0$,

$$v(\rho) = a_0 + p(R - H + 1) \times \left[ \frac{(Hc_2 + c_1\sqrt{-D})\sin\left( \frac{\sqrt{-D}}{2}\rho \right) + (Hc_1 - c_2\sqrt{-D})\cos\left( \frac{\sqrt{-D}}{2}\rho \right)}{((H-2)c_2 + c_1\sqrt{-D})\sin\left( \frac{\sqrt{-D}}{2}\rho \right) + ((H-2)c_1 - c_2\sqrt{-D})\cos\left( \frac{\sqrt{-D}}{2}\rho \right)} \right]. \tag{43}$$

Again, using the wave transformation from Equation (37), the other soliton solution of Equation (3) is written as

$$v(x, t) = a_0 + p(R - H + 1) \times$$
$$\left[ \frac{(Hc_2 + c_1\sqrt{-D})\sin\left( \frac{\sqrt{-D}}{2}\left[px - \frac{\eta}{\beta} t^\beta\right] \right) + (Hc_1 - c_2\sqrt{-D})\cos\left( \frac{\sqrt{-D}}{2}\left[px - \frac{\eta}{\beta} t^\beta\right] \right)}{((H-2)c_2 + c_1\sqrt{-D})\sin\left( \frac{\sqrt{-D}}{2}\left[px - \frac{\eta}{\beta} t^\beta\right] \right) + ((H-2)c_1 - c_2\sqrt{-D})\cos\left( \frac{\sqrt{-D}}{2}\left[px - \frac{\eta}{\beta} t^\beta\right] \right)} \right], \tag{44}$$

where $\eta = -p + 2p^2 R + 2pa_0 - p^2 H$.

## 5. Interpretation of the Numerical Outputs along with Their Graphical Portraiture

In this graphical section, we want to figure out some soliton solutions of the time-fractional nonlinear PDEs which are investigated in this article. This article aims to solve the KGE, the STOE, and the CRWPE in the sense of the conformable fractional derivative by applying the expansion method. The results obtained in this research paper are consistent, ordinary, and more accurate than the results of all previous papers. The accuracy and stability were looked over by setting the attained results back into the main equations, and we found exactness. The proposed expansion method gives two types of solitary-wave solutions: one is the exponential-function solution, and the other is the trigonometric-

function solution. The above Equations (22), (30), (34), and (42) represent the exponential-function solutions, and Equations (24), (32), (36), and (44) represent trigonometric-function solutions. For simplicity of the article, the graphical interpretations of Equations (34), (36), and (44) are omitted. The data in the graph displays the characteristics of the traveling wave solutions by inserting various parametric values.

Figures 1–5 exhibits the exponential and trigonometric solitary-wave solutions of the mentioned three model equations in the shape of a 3D plot (for $\beta = 0.5$), contour plot (for $\beta = 0.5$), and 2D plot (for $\beta = 0.25, 0.5, 0.75$). Figure 1 implies the smooth kink-shaped soliton solution with the parameters as $a_0 = 1$, $H = 1$, $R = 0.1$, $c_1 = 1$, $c_2 = 1$, $m = 1$, $\mu = 0.01$, and $T = 0.6$ in the case of the 3D plot (for $-100 \leq x \leq 100$ and $0 \leq t \leq 100$), contour plot (for $-100 \leq x \leq 100$ and $0 \leq t \leq 100$), and 2D plot (for $x = 0$ and $0 \leq t \leq 10$). These figure shows that when increasing the order $\beta$, the slope of the graph is increasing.

Again, Figure 2 implies the multiple singular periodic solution with the parameters as $a_0 = 1$, $H = 2$, $R = 3$, $c_1 = 1$, $c_2 = 1$, $m = 1$, $\mu = 3$ and $T = -8$ in the case of the 3D plot (for $-5 \leq x \leq 5$ and $0 \leq t \leq 5$), contour plot (for $-5 \leq x \leq 5$ and $0 \leq t \leq 5$), and 2D plot (for $x = 0$ and $0 \leq t \leq 0.5$). These figure shows that when increasing the order $\beta$, the curve scatters.

Figure 3 implies the anti-kink soliton solution with the parameters as $a_0 = 1$, $H = 3$, $R = 1.5$, $c_1 = 1$, $c_2 = -1$, $d = 1$, $\mu = 3.75$, and $T = 3$ in the case of the 3D plot (for $-100 \leq x \leq 100$ and $0 \leq t \leq 100$), contour plot (for $-100 \leq x \leq 100$ and $0 \leq t \leq 100$), and 2D plot (for $x = 0$ and $0 \leq t \leq 3$). These figure shows that when increasing the order $\beta$, the graph is shifted right.

Figure 4 implies the singular periodic solution with the parameters as $a_0 = 1$, $H = 1$, $R = 2.5$, $c_1 = -1$, $c_2 = 1$, $d = -0.01$, $\mu = -0.2475$ and $T = -9$ in the case of the 3D plot (for $-3 \leq x \leq 3$ and $0 \leq t \leq 3$), contour plot (for $-3 \leq x \leq 3$ and $0 \leq t \leq 3$), and 2D plot (for $x = 0$ and $0 \leq t \leq 50$). These figure shows that when increasing the order $\beta$, the graph closes together.

Figure 5 implies the bright kink-shaped soliton solution with the parameters as $a_0 = 1$, $H = 1$, $R = 0.2$, $c_1 = 1$, $c_2 = 1$, $d = 0.1$, $p = -3$, $\eta = -8.4$ and $T = 0.2$ in the case of the 3D plot (for $-100 \leq x \leq 100$ and $0 \leq t \leq 100$), contour plot (for $-100 \leq x \leq 100$ and $0 \leq t \leq 100$), and 2D plot (for $x = 0$ and $0 \leq t \leq 5$). These figure show that when increasing the order $\beta$, the graph is shifted right.

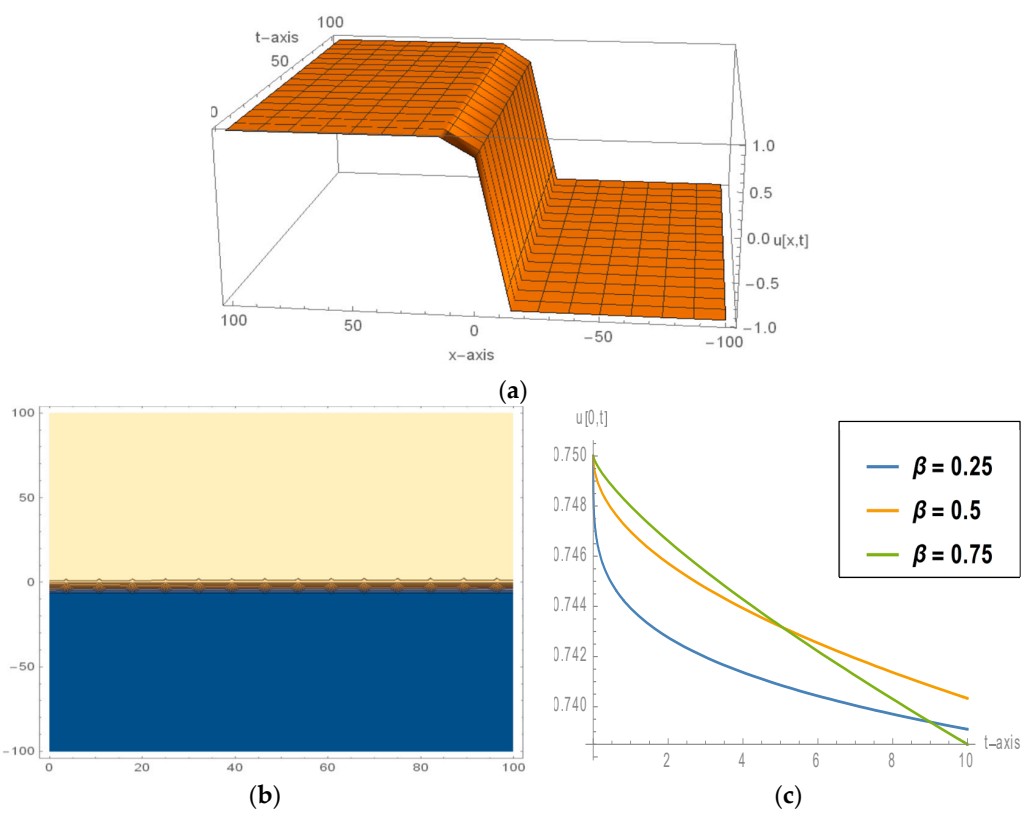

**Figure 1.** The smooth kink-shaped soliton solution for Equation (22) with the imposed values: (**a**) 3D plot, (**b**) contour plot, and (**c**) 2D plot for *β* = **0.25, 0.5, 0.75**.

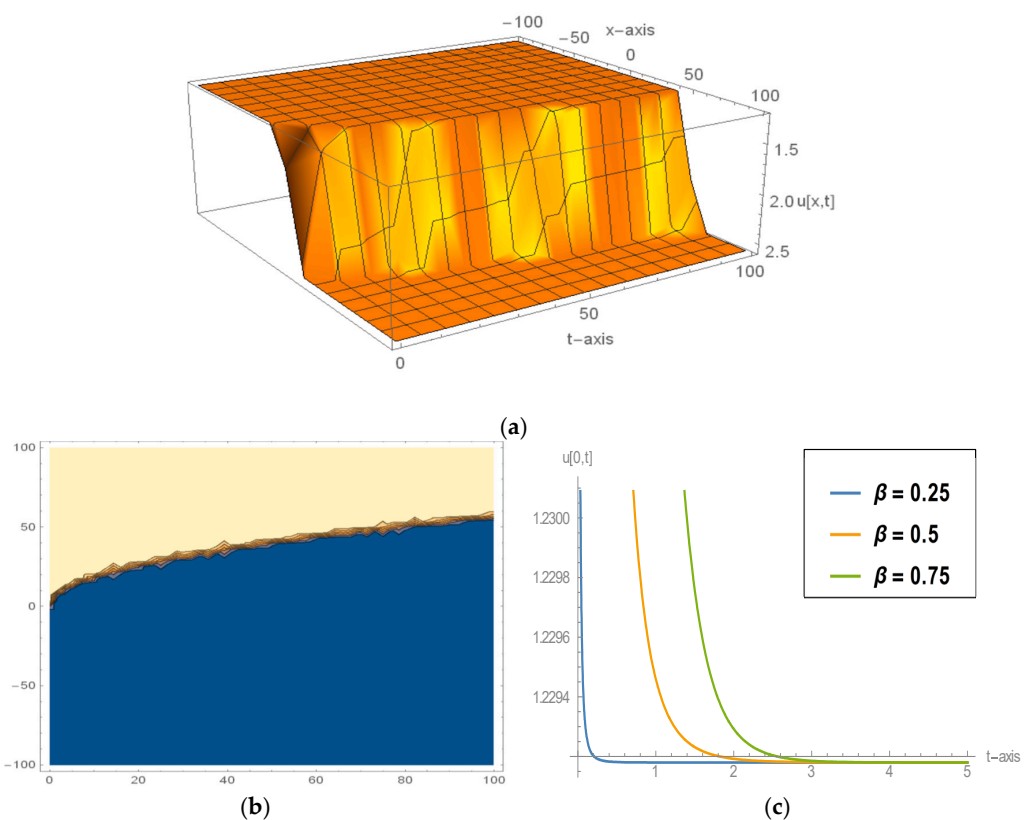

**Figure 2.** The multiple singular periodic solution for Equation (24) with the imposed values: (**a**) 3D plot, (**b**) contour plot, and (**c**) 2D plot for *β* = **0.25, 0.5, 0.75**.

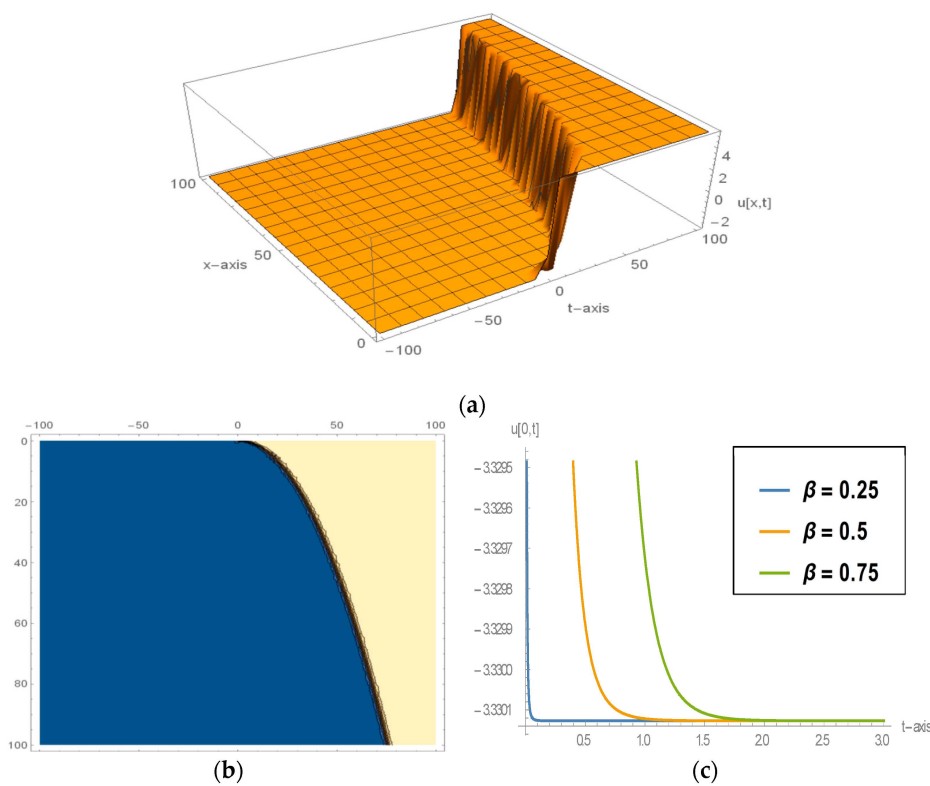

**Figure 3.** The anti-kink-shaped soliton solution for Equation (30) with the imposed values: (**a**) 3D plot, (**b**) contour plot, and (**c**) 2D plot for $\beta$ = 0.25, 0.5, 0.75.

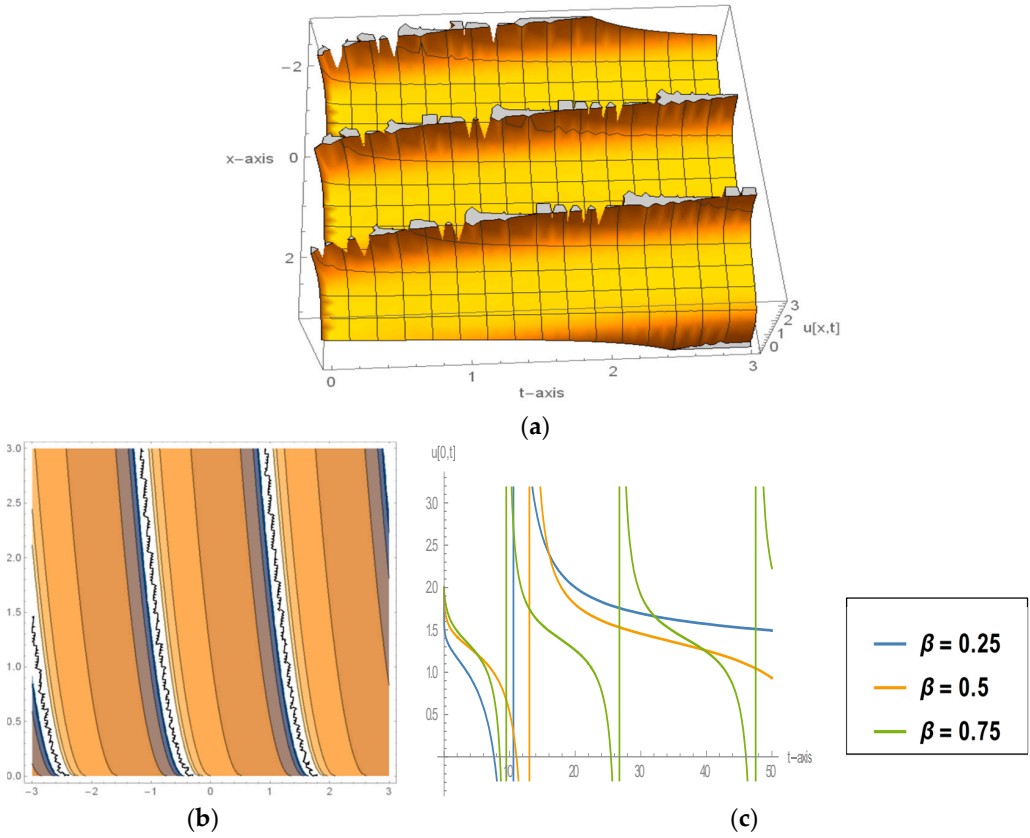

**Figure 4.** The singular periodic solution for Equation (32) with the imposed values: (**a**) 3D plot, (**b**) contour plot, and (**c**) 2D plot for $\beta$ = 0.25, 0.5, 0.75.

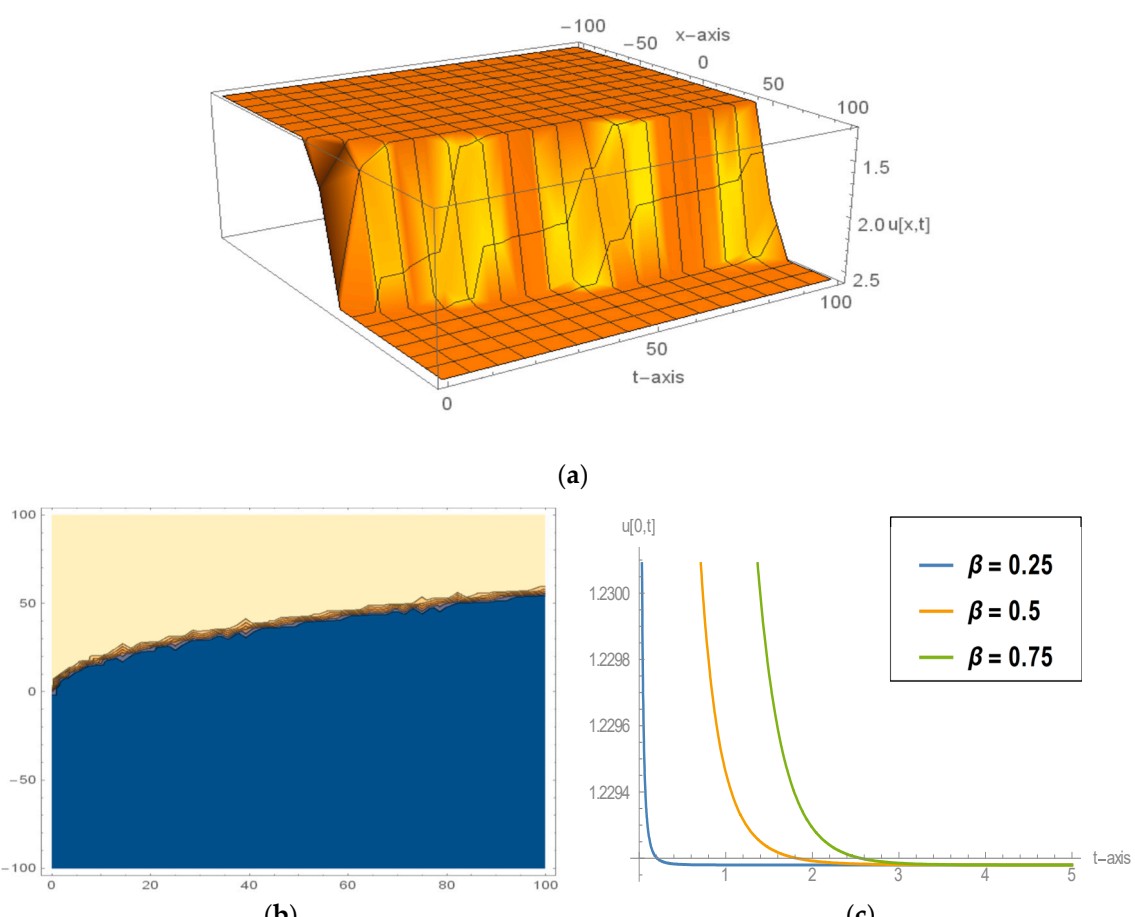

**Figure 5.** The bright kink-shaped soliton solution for Equation (42) with the imposed values: (**a**) 3D plot, (**b**) contour plot, and (**c**) 2D plot for $\beta$ = 0.25, 0.5, 0.75.

From the overall outcomes, it is understood that the fractional-order derivative reforms the nature of the results, narrates the continuous behavior of the waveform and has a remarkable influence on the nonlinear propagation related to the soliton solutions.

## 6. Comparison of the Results

To investigate the novelty of this work, we compared the findings we obtained with the previously obtained results of the time-fractional KGE [44], the time-fractional STOE [44], and the time-fractional CRWPE [48] in Tables 1–3, respectively.

**Table 1.** Comparison (I).

| Comparison of the obtained results with those of Taghizadeh et al. [44] | |
|---|---|
| Results of Taghizadeh et al. [44] using the simplest equation method on time-fractional KGE | Results obtained using the $\left(\frac{G'}{G'+G+A}\right)$-expansion method |
| For $a_0 = \pm\frac{\sqrt{-\theta_1\theta_2}}{\theta_1}$, $b = \pm\sqrt{\frac{\theta_2}{2(\lambda^2-l^2)}}$, $a = \pm\sqrt{\frac{2\theta_2}{(l^2-\lambda^2)}}$, $a > 0$ and $b < 0$, then the result $(i)$ gives exponential function solution as $$u(x,\,t) = \pm\left(\sqrt{-\frac{\theta_1}{\theta_2}} - \frac{\sqrt{\theta_2}}{\sqrt{2(\lambda^2-l^2)}}\frac{a_1 exp\left[\frac{\sqrt{2\theta_2}}{\sqrt{(\lambda^2-l^2)}}\left(lx-\frac{\lambda}{\Gamma(1+\alpha)}t^\alpha+\xi_0\right)\right]}{1-a_1\frac{\sqrt{\theta_2}}{\sqrt{2(\lambda^2-l^2)}}exp\left[\frac{\sqrt{2\theta_2}}{\sqrt{(\lambda^2-l^2)}}\left(lx-\frac{\lambda}{\Gamma(1+\alpha)}t^\alpha+\xi_0\right)\right]}\right).$$ Again for $a_0 = \pm\frac{\sqrt{-\theta_1}}{\theta_2}$, $A = -\frac{\sqrt{2\theta_2}}{(l^2-\lambda^2)}$, then the result $(ii)$ gives hyperbolic function results as $$u(x,\,t) = \pm\frac{\sqrt{-\theta_1}}{\theta_2}\times tanh\left(\sqrt{\frac{\theta_1}{2(l^2-\lambda^2)}}\left(lx-\frac{\lambda}{\Gamma(1+\alpha)}t^\alpha+\xi_0\right)\right).$$ | For $a_0 = a_0$, $a_1 = \frac{2a_0(H-R-1)}{2R-H}$, $m = m$, $\mu = \mu$, $k_1 = \frac{1}{2}m^2H^2 - \frac{1}{2}\mu^2H^2 + 2\mu^2R - 2m^2R$, $k_2 = \frac{(H-2R)^2(\mu^2-m^2)}{2a_0^2}$, then result $(i)$ $D = H^2 - 4R > 0$ gives exponential function solution as $$v(x,\,t) = a_0 + \frac{2a_0(H-R-1)}{2R-H}\times\left[\frac{c_1(H+\sqrt{D})+c_2(H-\sqrt{D})e^{\sqrt{D}[mx-\frac{\mu}{\beta}t^\beta]}}{c_1(H+\sqrt{D}-2)+c_2(H-\sqrt{D}-2)e^{\sqrt{D}[mx-\frac{\mu}{\beta}t^\beta]}}\right].$$ $(ii)$ $D = H^2 - 4R < 0$ gives trigonometric function solution as $$v(x,\,t) = a_0 + \frac{2a_0(H-R-1)}{2R-H}\times$$ $$\left[\frac{(Hc_2+c_1\sqrt{-D})sin\left(\frac{\sqrt{-D}}{2}[mx-\frac{\mu}{\beta}t^\beta]\right)+(Hc_1-c_2\sqrt{-D})cos\left(\frac{\sqrt{-D}}{2}[mx-\frac{\mu}{\beta}t^\beta]\right)}{((H-2)c_2+c_1\sqrt{-D})sin\left(\frac{\sqrt{-D}}{2}[mx-\frac{\mu}{\beta}t^\beta]\right)+((H-2)c_1-c_2\sqrt{-D})cos\left(\frac{\sqrt{-D}}{2}[mx-\frac{\mu}{\beta}t^\beta]\right)}\right].$$ |

**Table 2.** Comparison (II).

| Comparison of the obtained results with those of Taghizadeh et al. [44] | |
|---|---|
| Results of Taghizadeh et al. [44] using the simplest equation method on time-fractional STOE | Results obtained using the $\left(\frac{G'}{G'+G+A}\right)$-expansion method |
| For $a = -a_0$, $b = -\frac{a_1}{2}$, $R = 0$, $\lambda = ka_0^2$  $a > 0$ and $b < 0$, then the result $(i)$ gives the exponential function solution as $$u(x,\,t) = a_0\left(\frac{2-a_1 exp\left[-a_0\left(x-\frac{ka_0^2}{\Gamma(1+\alpha)}t^\alpha+\xi_0\right)\right]}{2+a_1 exp\left[-a_0\left(x-\frac{ka_0^2}{\Gamma(1+\alpha)}t^\alpha+\xi_0\right)\right]}\right).$$ For $A = -a_0$, $a_1 = 1$, $r = 0$, $\lambda = ka_0^2$, $a_1 = 2$, then the result $(ii)$ gives hyperbolic function solution as $$u(x,\,t) = -a_0\,tanh\left(\left(x-\frac{ka_0^2}{\Gamma(1+\alpha)}t^\alpha+\xi_0\right)\right).$$ | For $a_0 = H - 2R$, $a_1 = 2R - 2H + 2$, $N = 0$, $d = d$, $\mu = -4dR + dH^2$, then result $(i)$ $D = H^2 - 4R > 0$ gives exponential function solution as $$v(x,\,t) =$$ $$a_0 + (R-H+1)\times\left[\frac{c_1(H+\sqrt{D})+c_2(H-\sqrt{D})e^{\sqrt{D}[x-\frac{\mu}{\beta}t^\beta]}}{c_1(H+\sqrt{D}-2)+c_2(H-\sqrt{D}-2)e^{\sqrt{D}[x-\frac{\mu}{\beta}t^\beta]}}\right],$$ where $$\mu = -3dHR - dR + 6da_0R + 3dR^2 + 3da_0^2 - 3da_0H + dH^2.$$ $(ii)$ $D = H^2 - 4R < 0$ gives trigonometric function solution as $$v(x,\,t) = a_0 + (R-H+1)\times$$ $$\left[\frac{(Hc_2+c_1\sqrt{-D})sin\left(\frac{\sqrt{-D}}{2}\rho\right)+(Hc_1-c_2\sqrt{-D})cos\left(\left(\frac{\sqrt{-D}}{2}\rho\right)\right)}{((H-2)c_2+c_1\sqrt{-D})sin\left(\left(\frac{\sqrt{-D}}{2}\rho\right)\right)+((H-2)c_1-c_2\sqrt{-D})cos\left(\left(\frac{\sqrt{-D}}{2}\rho\right)\right)}\right],$$ where $$\mu = -3dHR - dR + 6da_0R + 3dR^2 + 3da_0^2 - 3da_0H + dH^2 \text{ and}$$ $$\rho = \left(x - \frac{\mu}{\beta}t^\beta\right).$$ |

**Table 3. Comparison (III).**

| Comparison of the obtained results with the results of Guner et al. [48] | |
| --- | --- |
| Results of Guner et al. [48] using $\left(\frac{G'}{G}\right)$-expansion method on time-fractional CRWPE | Results obtained using the $\left(\frac{G'}{G'+G+A}\right)$-expansion method |

For $a_0 = a_0$, $a_1 = k$, $c = -k^2\lambda + 2a_0 k + k$, $k = k$,

$\xi_0 = -a_0 k^2 \lambda + ka_0^2 + k^3\mu$, $C_1 \neq 0$, $C_2 = 0$, $\lambda > 0$ and $\mu = 0$, then

(i) $\lambda^2 - 4\mu > 0$ gives hyperbolic solution as

$$U_1(x,\ t) = a_0 - \frac{k\lambda}{2} + \frac{k\lambda}{2} coth\left\{\frac{\lambda}{2}\frac{(kx - k - k^2\lambda + 2a_0)t^\alpha}{\Gamma(1+\beta)}\right\}.$$

(ii) $\lambda^2 - 4\mu < 0$ gives rational function solution as

$$U_2(x,\ t) = a_0 - \frac{k\lambda}{2} + \frac{k\sqrt{4\mu-\lambda^2}}{2}\left(\frac{-C_1 \sin\frac{1}{2}\sqrt{4\mu-\lambda^2}\xi + C_2\cos\frac{1}{2}\sqrt{4\mu-\lambda^2}\xi}{C_1\cos\frac{1}{2}\sqrt{4\mu-\lambda^2}\xi + C_2\sin\frac{1}{2}\sqrt{4\mu-\lambda^2}\xi}\right),$$

where $\xi = \frac{(kx - k - k^2\lambda + 2a_0)t^\alpha}{\Gamma(1+\beta)}$.

For $a_0 = a_0$, $a_1 = p(R - H + 1)$, $p = p$, $\eta = -p + 2p^2 R + 2pa_0 - p^2 H$,

$\rho_0 = 2a_0 p^2 R + pa_0^2 - a_0 p^2 H + p^3 R^2 - p^3 HR + p^3 R$, then

(i) $D = H^2 - 4R > 0$ gives exponential function solution as

$$v(x,\ t) = a_0 + p(R - H + 1) \times \left[\frac{c_1(H+\sqrt{D}) + c_2(H-\sqrt{D})e^{\sqrt{D}[px - \frac{\eta}{\beta}t^\beta]}}{c_1(H+\sqrt{D}-2) + c_2(H-\sqrt{D}-2)e^{\sqrt{D}[px - \frac{\eta}{\beta}t^\beta]}}\right],$$

where $\eta = -p + 2p^2 R + 2pa_0 - p^2 H$.

(ii) $D = H^2 - 4R < 0$ gives trigonometric function solution as

$$v(x,\ t) = a_0 + p(R - H + 1) \times$$
$$\left[\frac{(Hc_2 + c_1\sqrt{-D})\sin\left(\frac{\sqrt{-D}}{2}\left[px - \frac{\eta}{\beta}t^\beta\right]\right) + (Hc_1 - c_2\sqrt{-D})\cos\left(\frac{\sqrt{-D}}{2}\left[px - \frac{\eta}{\beta}t^\beta\right]\right)}{((H-2)c_2 + c_1\sqrt{-D})\sin\left(\frac{\sqrt{-D}}{2}\left[px - \frac{\eta}{\beta}t^\beta\right]\right) + ((H-2)c_1 - c_2\sqrt{-D})\cos\left(\frac{\sqrt{-D}}{2}\left[px - \frac{\eta}{\beta}t^\beta\right]\right)}\right],$$

where $\eta = -p + 2p^2 R + 2pa_0 - p^2 H$.

## 7. Conclusions

Our research article focused on time-fractional PDEs via an effective computational technique. We obtained eight new soliton solutions for these models. In this paper, for the time-fractional KGE we obtained two new soliton solutions, for the time-fractional STOE we obtained four new soliton solutions, and for the time-fractional CRWPE, we obtained two new soliton solutions. Our obtained soliton solutions were classified into different forms of exponential and trigonometric functions. So, by using these extracted new solitons, we could easily explain the new physical phenomena of these mentioned models. By investigating these results, we could decide they played an influential role in providing models and physical configurations of many complicated natural occurrences and many dynamical systems. We chose some convenient values of the parameters for drawing graphs and ascertained 3D figures, 2D figures, and contour figures of the exciting shapes, such as a smooth kink-shaped soliton, an ant-kink-shaped soliton, a singular periodic solution, a multiple singular periodic solution, a bright kink-shaped soliton, etc. The mentioned method can be expected to play a vital role in solving various nonlinear fractional PDEs, especially the time-fractional nonlinear PDEs, as seen from our time-fractional model equations presented in our paper, as well as in [58]. It is important to emphasize that no research study has achieved reliable and instructive results similar to those found here.

**Author Contributions:** Conceptualization, M.A.I. and M.M.M.; methodology, M.A.I.; software, M.S.O.; validation, M.A.I., A.H.G. and M.M.M.; formal analysis, M.S.O.; investigation, M.M.M.; resources, A.H.G.; data curation, M.A.I.; writing—original draft preparation, M.S.O.; writing—review and editing, M.M.M.; visualization, M.M.M.; supervision, M.S.O.; project administration, M.S.O.; funding acquisition, A.H.G. All authors have read and agreed to the published version of the manuscript.

**Funding:** This research received no external funding.

**Data Availability Statement:** The data supporting this study's findings are available from the corresponding author upon reasonable request.

**Conflicts of Interest:** The authors declare that they have no conflict of interest.

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
