# Peer review of "Extracting the Ultimate New Soliton Solutions of Some Nonlinear Time Fractional PDEs via the Conformable Fractional Derivative"

_fractalfract, doi:10.3390/fractalfract8040210_

Round 1
Reviewer 1 Report
The authors have investigated the fractional order partial differential equations such as the time-fractional Klein-Gordon equation (KGE), the time-fractional Sharma-Tasso-Olever equation (STOE), and the time-fractional Clannish Random Walker’s Parabolic equation (CRWPE) by using an expansion method for extracting new soliton solutions. The obtained results are interesting. However, I have the following suggestions:
1. Page 1, line 3: the conformable fractional derivatives can be changed conformable fractional derivative.
2. The need to shorten the abstract of the paper.
3. Page 5, line 178: The title of section 4 is missing. The authors need to add the title in section 4.
4. Page 6, line 218: The alignment required in equation (4.9).
5. The authors at least compare the obtained result with the existing literature if possible.
6. Is this method applicable to initial and boundary value problems? If yes, the authors should add some lines about the the initial and boundary value problems in the conclusion. It may be an additional advantage of the discussed method.
After minor revision, I recommend it for the publication in the esteemed journal.
Author Response
Reviewer #1:
The authors have investigated the fractional order partial differential equations such as the time-fractional Klein-Gordon equation (KGE), the time-fractional Sharma-Tasso-Olever equation (STOE), and the time-fractional Clannish Random Walker’s Parabolic equation (CRWPE) by using an expansion method for extracting new soliton solutions. The obtained results are interesting. However, I have the following suggestions:
- Page 1, line 3: the conformable fractional derivatives can be changed conformable fractional derivative.
Reply: According to the suggestion by the reviewer, we have modified it.
- The need to shorten the abstract of the paper.
Reply: We have shortened and enriched the abstract as per the reviewer’s suggestion.
- Page 5, line 178: The title of section 4 is missing. The authors need to add the title in section 4.
Reply: According to the suggestion by the reviewer, we have added the title in section 4, and the give title is “Application of the proposed three model equations”.
- Page 6, line 218: The alignment required in equation (4.9).
Reply: According to the suggestion by the reviewer, we have aligned the equation (4.9) on page no.6.
- The authors at least compare the obtained result with the existing literature if possible.
Reply: According to the suggestion by the reviewer, we have compared our obtained results with the previous results.
- Is this method applicable to initial and boundary value problems? If yes, the authors should add some lines about the initial and boundary value problems in the conclusion. It may be an additional advantage of the discussed method.
Reply: To the best of our knowledge, no one used this method in solving initial and boundary value problems. We will try to do it in future work.
After minor revision, I recommend it for publication in the esteemed journal.

Reviewer 2 Report
1. The written manuscript looks shifted to the right that affect lines of 244,247,251,254,257,286, 289, and so on. Different fonts are seen in the whole manuscript.
So, format should be modified.
2. Is the method extendable to coupled systems of differential equations? If so, how? These coupled systems with soliton solutions are seen in theoretical physics problems. So, the community will benefit.
3. In sec 5, authors claim that their results are more accurate than previous results. So, it is expected to see a comparison at least in a table form. Nothing has been cited in this regard.
4. If something has been omitted for simplicity should be presented in Appendix.
5. Figures are at their unqualified status. All, should be provided with high resolution.
6. Figures are presented, however no related discussion is seen.
7. Section 5 should be formatted in a way that readers can clearly see the results in various paragraph.
8. That would be better to present some data in table regarding the accuracy and stability including various equations. So, each time readers do not need to go to the first page to see the equations.
The English look well written. However, in the final version should be check carefully again.
Author Response
Reviewer #2:
Comments and Suggestions for Authors:
- The written manuscript looks shifted to the right that affect lines of 244,247,251,254,257,286, 289, and so on. Different fonts are seen in the whole manuscript. So, format should be modified.
Reply: Sorry for our mistakes. We corrected in our revised manuscript. Please check and confirm.
- Is the method extendable to coupled systems of differential equations? If so, how? These coupled systems with soliton solutions are seen in theoretical physics problems. So, the community will benefit.
Reply: Yes, our method can be extended to solve coupled systems of differential equations. We will study this kind of model in future work.
- In sec 5, authors claim that their results are more accurate than previous results. So, it is expected to see a comparison at least in table form. Nothing has been cited in this regard.
Reply: According to the suggestion by the reviewer, we have compared our obtained results with the previous results in tables 1-3.
- If something has been omitted for simplicity should be presented in Appendix.
Reply: Nothing is omitted, Thank you.
- Figures are at their unqualified status. All should be provided with high resolution.
Reply: Thank you. All the figures have been updated and corrected.
- Figures are presented, however no related discussion is seen.
Reply: As per reviewer’s suggestion, we have given a brief discussion of the curves according to the variation of fractional order.
- Section 5 should be formatted in a way that readers can clearly see the results in various paragraph.
Reply: As per reviewer’s suggestion, to better understand the results mentioned in Section 5, it have written in the form of subsections.
- That would be better to present some data in table regarding the accuracy and stability including various equations. So, each time readers do not need to go to the first page to see the equations.
Reply: According to the suggestion by the reviewer, we have compared our obtained results with the previous results including various equations.
Comments on the Quality of English Language
The English look well written. However, in the final version should be check carefully again.
Reply: The whole paper is revised.

Reviewer 3 Report
see attachment

The quality of English language can be improved
Author Response
Reviewer #3:
In this article, the authors investigate the time fractional PDEs via an effective computational technique given in methodology and obtained eight new soliton solutions of these nonlinear fractional order PDEs. For time fractional KGE, two new soliton solutions are obtained, for time fractional STOE our new soliton solutions are obtained, and for time fractional CRWPE, two new soliton solutions are obtained.
I suggest a major revision. Some problems are as follows.
Point 1: The abstract should be revised. Generally, the abstract should be short and overarching. Reply: According to the suggestion by the reviewer, we have shortened and enriched the abstract.
Point 2: The major innovation point should be referred.
Reply: Thank you. Done.
Point 3: The introduction is messy. The authors refer “Moreover, the application of the Fourier spectral approach in the procedure of getting solutions to the space non-integer order reaction diffusion can be noticed in [11, 12].” For spectral approach for non-integer order reaction diffusion situation, the authors can refer: a space-time spectral order sine-collocation method for the fourth-order nonlocal heat model arising in viscoelasticity. A Predictor-Corrector Compact Difference Scheme for a nonlinear Fractional Differential Equation.
Reply: According to the suggestion by the reviewer, we have inserted the articles, “A space-time spectral order sine-collocation method for the fourth-order nonlocal heat model arising in viscoelasticity” into reference no. [11] and “A predictor-corrector compact difference scheme for a nonlinear fractional differential equation” into reference no. [12].
Point 4: Page 5, “4.
4.1. Investigation of the KGE”, the title should be added.
Reply: As per reviewer’s suggestion, we have added the title in Section 4, and the given title is “Application”.
Point 5: Page 6, Eq. (4.6) and (4.7) should be arranged. Such as (4.26), (4.27).
Reply: Thanks for your deep observation. It’s our mistake. We corrected our revised manuscript.
Point 6: Fig.1(c), Fig.2(c) should be arranged.
The problem formulation section needs more attention. Describe the parameters and transmission in a clear way.
Reply: Now, we have done it. Please check.
Point 7: Some words have extra space between them.
Reply: We already done it. Please check.
Point 8: It is suggested to comma in equation.
Reply: We have done it and thanks for your help.
Point 9: There may be problems with the format of the reference, and it is recommended to check it.
Reply: We check very carefully. Hope now its fine. Please check.
Point 10: The section about conclusion, the authors refer “The mention method can be expected to play a vital role in solving various nonlinear PDEs, especially the time fractional nonlinear PDEs, as seen from our time fractional model equations represented in our paper”. I suggest a reference: an implicit robust numerical scheme with graded meshes for the modified Burgers model with nonlocal dynamic properties.
Reply: According to the suggestion by the reviewer, we have inserted the article, “An implicit robust numerical scheme with graded meshes for the modified Burgers model with nonlocal dynamic properties” into reference no. [58].

Round 2
Reviewer 3 Report
Thanks the authors for taking my comments into consideration, as one can see, this revised version has been improved greatly, thus, the manuscript can be accepted.
it is ok